# Effect of High-Intensity Interval Training vs. Moderate-Intensity Continuous Training on Fat Loss and Cardiorespiratory Fitness in the Young and Middle-Aged a Systematic Review and Meta-Analysis

**DOI:** 10.3390/ijerph20064741

**Published:** 2023-03-08

**Authors:** Zhicheng Guo, Meng Li, Jianguang Cai, Weiqi Gong, Yin Liu, Ze Liu

**Affiliations:** School of Physical Education, Hunan University of Science and Technology, Xiangtan 411201, China

**Keywords:** high-intensity interval training, moderate-intensity continuous training, young, middle-aged, fat loss, cardiorespiratory fitness, systematic review

## Abstract

Objectives: This systematic review is conducted to evaluate the effect of high-intensity interval training (HIIT) and moderate-intensity continuous training (MICT) on body composition and cardiorespiratory fitness (CRF) in the young and middle-aged. Methods: Seven databases were searched from their inception to 22 October 2022 for studies (randomized controlled trials only) with HIIT and MICT intervention. Meta-analysis was carried out for within-group (pre-intervention vs. post-intervention) and between-group (HIIT vs. MICT) comparisons for change in body mass (BM), body mass index (BMI), waist circumference (WC), percent fat mass (PFM), fat mass (FM), fat-free mass (FFM), and CRF. Results: A total of 1738 studies were retrieved from the database, and 29 studies were included in the meta-analysis. Within-group analyses indicated that both HIIT and MICT can bring significant improvement in body composition and CRF, except for FFM. Between-group analyses found that compared to MICT, HIIT brings significant benefits to WC, PFM, and VO_2peak_. Conclusions: The effect of HIIT on fat loss and CRF in the young and middle-aged is similar to or better than MICT, which might be influenced by age (18–45 years), complications (obesity), duration (>6 weeks), frequency, and HIIT interval. Despite the clinical significance of the improvement being limited, HIIT appears to be more time-saving and enjoyable than MICT.

## 1. Introduction

The population of overweight and obese individuals has increased relentlessly for almost 40 years [1]. Excess weight and obesity are issues for close to 30% of the population worldwide [2]. Metabolic dysfunction, Type 2 diabetes, cardiovascular disease (CVD), and other health complications are associated with an uncontrollable increase in weight [3,4]. Worse, being obese or overweight for an extended period not only creates difficulties in daily life but also leads to an elevated risk of depression and anxiety due to societal discrimination and prejudice against individuals with excess weight [5,6].

For obesity, the focus should be on reducing body fat rather than just achieving a restricted amount of weight loss [4,7]. To date, diet intervention [8], lifestyle modification [9], exercise [10], drugs [11], and the combination of the above [12,13,14] have been systematically applied to weight loss. However, numerous guidelines focusing only on weight loss, the significance of fat-free mass (FFM), and cardiorespiratory fitness (CRF) in overall fitness were often overlooked [15]. A loss of FFM means a decrease in basal metabolic rate, which is unconducive to fat loss [16]. In addition, a high level of CRF brings all-cause mortality risk down and improves physical activity levels [17].

It has been widely proven that exercise has a positive effect on fat loss and CRF [18,19]. In most exercise prescriptions for fat loss, moderate-intensity continuous training (MICT) is the first choice (sometimes combined with resistance training) and the same finding is also true for CRF improvement [20,21]. However, in order to achieve such a benefit, MICT needs to be maintained for a long time (≥150 min/day or 1000 min/week), which is a high time cost for young and middle-aged people today [22,23]. High-intensity interval training (HIIT) has been regarded as a famous fitness trend for its time-saving and effective characteristics in recent years [24]. Several randomized controlled trials (RCTs) were conducted to explore if HIIT is a better exercise form than MICT for fat loss and CRT improvement [25,26,27].

Compared to MICT, HIIT has similar or better influences on ventricular and endothelial function [28] and peak rate of oxygen consumption (VO_2peak_) [29]. For Type 2 diabetes and hypertension, HIIT is indicated to be equivalent to or greater than MICT in reducing insulin resistance [30] and blood pressure [31]. While the training volume in HIIT is less than that of MICT, both HIIT and MICT have similar benefits in skeletal muscle [32], exercise adaptation, and exercise performance [33]. In addition, participants allocated to the HIIT programs, compared with MICT, have a higher level of enjoyment and adherence [34], which profited from its various designs.

The youth and middle-aged often face a common obstacle to engaging in physical activity, which is the insufficient amount of time available [35]. Over the last decade, several studies took adolescents or the elderly as the core subjects in exploring the effect of HIIT intervention, however, these studies ignored the biggest beneficiaries of HIIT timeliness—the young and middle-aged. In addition, as core measures of the effects of exercise, the improvement in body composition and CRF are inseparable and need to be explored together. Although there is ample evidence indicating that HIIT is more advantageous for body composition and CRF than MICT for all age groups, there is still no consensus on whether HIIT is as effective or more effective than MICT in terms of fat loss and CRF in the young and middle-aged. This systematic review was conducted (1) to compare the impact of HIIT and MICT on fat loss and CRF in the young and middle-aged; (2) to determine the suitable intervention population for HIIT and the more effective forms of HIIT on the young and middle-aged.

## 2. Materials and Methods

This systematic review was registered in the International Prospective Register of Systematic Reviews (PROSPERO). The registration number was CRD42022330406.

### 2.1. Literature Search Strategy

The review was conducted following the guidelines of the PRISMA-P statement [36]. A throughout search of the electronic literature was carried out up to 22 October 2022, including Pubmed, Embace, the Cochrane Library, Web of Science, CNKI, CBM, and Wanfang. The search criteria were developed through some similar systematic reviews [37,38], ‘high-intensity interval training’, ‘moderate-intensity continuous training’, and so on were chosen as the key phrases. Search results were imported into a reference manager (Endnote X9). To make sure more relevant studies were included in the review, we also examined the reference lists of the eligible studies. The papers were appraised by two researchers (Guo and Gong) independently. After conducting a comprehensive evaluation, the papers that met our criteria were included. Any disputes were solved by a third researcher (Cai) through conversation.

### 2.2. Inclusion and Exclusion Criteria

All the papers were screened following the criteria shown in Table 1. The following PICOS criteria were used in the screen:Participants

The mean age of the participants in eligible studies was between 18 and 60 years. The subjects were limited to human and animal-based, and age-incompatible subjects were excluded. There was no restriction on participants with medical comorbidities in this review, the health status of the included participants was indicated in the basic characteristics.

Intervention

The intervention of the participants was HIIT only. The intensity of HIIT was measured between 80–100% HRmax or VO_2peak_, or at maximum effort, or a rating of perceived exertion (RPE) greater than 15 [39]. The duration of HIIT was a minimum of 4 weeks. The passive recovery or low-intensity exercise in HIIT was between 30 s to 4 min. The review did not limit the form of HIIT, but HIIT combined with other interventions (e.g., resistance training) was excluded.

Comparison

The included studies comprised a comparator group that undertook MICT. The training programs of MICT were at intensity 40 to 80% HRmax or VO_2peak_, or an RPE between 12–15, and the duration was over 15 min. There was also no restriction on the form of MICT.

Outcomes

The primary data related to fat loss (body mass (BM), body mass index (BMI), waist circumference (WC), fat mass (FM), fat-free mass (FFM), percent fat mass (PFM)) and cardiorespiratory fitness (VO_2peak_, systolic blood pressure (SBP), diastolic blood pressure (DBP)) were included. The outcomes were all directly reported in the studies, the recalculated values were excluded.

Study

Research involving randomized controlled trials (RCT) written in English and Chinese. Observational studies, reviews, and studies and abstracts without adequate data were excluded. 

### 2.3. Data Extraction

Two researchers (Guo and Gong) extracted the data independently, and the extracted results were checked by another two researchers (Li and Liu). The basic characteristics of the studies including age, sex ratio, the health status of participants, form of intervention, participants’ population, intervention characteristics (intensity, duration, and frequency of HIIT and MICT), and dropouts (both HIIT and MICT) were extracted. The mean ± standard deviation values, mean difference (MD), and 95% confidence intervals (95% CI) of pre-intervention, post-intervention, and changes between pre- and post-intervention (if reported) were extracted. When the outcomes reported by the studies were insufficient or hard to extract, we would contact the corresponding authors for the data needed in the meta-analysis.

### 2.4. Study Quality Assessment

The appraisal of the included studies was conducted by two reviewers (Guo and Li). Considering that HIIT and MICT are forms of physiotherapy, a Physiotherapy Evidence Database (PEDro) scale [40] was used to assess the quality. The encompassing external validity (1 item), internal validity (8 items), and statistical reporting (2 items) of the eligible studies were checked to assess the quality. All items were rated yes or no according to whether the criterion is satisfied in the study.

### 2.5. Statistical Analysis

In this review, all analyses were carried out using the R package (V.4.1.2). The mean and standard deviation of the changes between pre-intervention and post-intervention in the HIIT and MICT groups were used to compare the between-group differences. If the mean and standard deviation of the changes were not reported directly, we would calculate them through pre-intervention and post-intervention values. Considering that some experimental endpoints were highly variable, we adopted a random-effects model for all outcomes. The MD was used to complete the effect size (ES) when outcome units in included studies were the same. If not, the standardized mean difference (SMD) with 95% CI would be used. The heterogeneity among studies was quantified using Cochran’s Q test and the inconsistency I^2^ test. When I^2^ was 0 to 50%, the heterogeneity was considered to be acceptable. Funnel plots and Egger’s test were adopted to assess the publication bias. To test the sensitivity, we carried out several subgroup analyses to find out whether the individual characteristics or intervention characteristics of each eligible study can influence the final result. Ages (18–45 and 45–60 years), complications (obesity (BMI > 30 kg/m^2^) and other chronic diseases), duration (≤6 and >6 weeks), frequency (≤3 and >3 times/week), and interval protocol (<3 and ≥3 min) were examined as subgroups.

## 3. Results

### 3.1. This Included Studies

As shown in Figure 1, the search strategy retrieved 1604 studies from electronic databases and 134 studies from references and other sources. After removing the duplicates, 1188 studies were evaluated via title and abstract, and 291 studies remained to be full-text screened. After diligently reviewing, 254 studies were removed for not meeting the inclusion criteria. Finally, 29 studies were evaluated as eligible and included in this analysis.

### 3.2. Participant and Intervention Characteristics

A total of 807 participants in 29 studies [41,42,43,44,45,46,47,48,49,50,51,52,53,54,55,56,57,58,59,60,61,62,63,64,65,66,67,68,69] were included in our meta-analysis. The primary characteristics of the participants and interventions are summarized in Table 2. The age of the participants was 33.82 ± 11.6 years. A total of 404 participants were allocated to the HIIT group, and 403 participants were allocated to the MICT group. A total of four studies [48,50,67,68] did not report sex ratio, the sex ratio of the remaining studies was 2:3. The participants in 20 included studies [44,45,46,47,48,49,51,52,53,54,55,57,58,59,60,61,63,66,68,69] were people with sedentary obesity, two studies [30,56] were sedentary only, and seven studies [41,42,43,50,62,64,67] were other medical comorbidities (two Type 1 diabetes, two Type 2 diabetes, one prediabetes, one polycystic ovary syndrome, and one fibromyalgia).

Out of all interventions, most exercise forms were cycling, six studies [47,49,50,63,66,69] used running, two studies [48,51] used home-based HIIT for HIIT and running for MICT, and one study [44] used boxing for HIIT and walking for MICT. The duration of 13 studies [44,46,47,49,51,53,54,55,58,63,66,67,69] was 3 months, only two studies [42,50] adopted >3 months intervention. Most interventions used HRmax or VO_2peak_ to measure the intensity of exercise, two studies [44,50] used RPE, three studies [45,59,60] used Wpeak or Wmax, six studies [46,48,52,56,57,61,64] used all-out exercise or maximum effort. A total of 10 HIIT interventions used passive recovery [44,48,51,53,57,58,59,61,64,69] and the remaining used active recovery. The exercise time ranged from 9 to 54 min for HIIT and 15 to 60 min for MICT, only two studies [58,68] used energy expenditure formulating exercise time. A total of 19 studies [42,43,46,47,48,50,51,52,53,54,56,57,59,60,61,62,64,67,69] instructed participants to exercise 3 times/week, nine studies [41,45,49,55,58,63,65,66,68] instructed > 3 times/week, and only one study [44] instructed once per week. A total of 14 studies [41,46,49,50,51,53,54,58,61,63,64,67,68,69] had dropouts, of which four studies [49,50,51,64] had <85% attendance rate.

### 3.3. Outcome Assessment

All studies measured body mass, BMI, and WC directly by using a digital scale, a stadiometer, and a plastic tape (WC was measured midway between the lowest rib and iliac crest in the horizontal plane). Out of the included studies that reported FM, FFM, and PFM, 10 studies [45,50,52,53,54,59,60,63,68,69] used dual-energy X-ray absorptiometry (DXA), seven studies [41,47,49,55,57,58,66] used bioelectrical impedance analysis (BIA), three studies [44,48,56] used six skinfold sites, one study [46] used air displacement plethysmography, and two studies [51,64] did not report the measuring method. Most studies adopted a graded maximal exercise test on an electronically braked cycle ergometer to measure VO_2peak_. Resting blood pressure was measured at the brachial artery manually or by an automated sphygmomanometer.

### 3.4. Quality Assessment

As shown in Appendix A, the quality of included studies was moderate (mean ± SD = 5.72 ± 0.83). Apart from two studies [49,51], all other studies randomly allocated participants, however, only four studies [42,44,47,67] used concealed allocation. A total of four studies [45,48,49,58] did not report and compare the baseline data of the participants. No studies blinded subjects or therapists due to the characteristics of exercise intervention. For assessors, most studies did not report the blind method and the remaining studies did not blind assessors. The overall attendance rate was 90.24%, six studies [49,50,51,63,64,67] lost more than 15% of participants to follow-up. All studies reported adequate information on intention, between-group statistics, and point measures.

### 3.5. Meta-Analysis

As summarized in Table 3, 28 studies reported pre- and post-intervention in BM, 25 studies reported BMI, 12 studies reported WC, 27 studies reported VO_2peak_, 14 studies reported SBP, and 14 studies reported DBP. Given PFM, FM, and FFM, one study did not report the baseline and follow-up. The within-group analyses were completed by using the above data. Significant heterogeneity(*p* < 0.01) was found for BM (MD: −2 kg for HIIT and −2.19 kg for MICT), BMI (MD: −0.9 kg/m^2^ for HIIT and −0.92 kg/m^2^ for MICT), WC (MD: −4.41 cm for HIIT and −2.96 cm for MICT), PFM (MD: −2.03% for HIIT and −1.89% for MICT), FM (MD: −1.79 kg for HIIT and −2.33 kg for MICT), VO_2peak_ (SMD: 0.83 for HIIT and 0.6 for MICT), and SBP (MD: −3.83 mmHg for HIIT and −3.56 mmHg for MICT). There was significant change (*p* < 0.05) in DBP (MD: −1.59 mmHg for HIIT and −1.88 mmHg for MICT). No significant differences were observed in the HIIT and MICT groups between baseline and follow−up in FFM (MD: −0.36 kg for HIIT and −0.38 kg for MICT).

Between-group analyses were performed using the change between pre-and post-intervention (Table 3). The analyses of HIIT vs. MICT on fat loss and cardiorespiratory fitness were presented in Figure 2. There were significant differences between HIIT and MICT in WC (MD = −0.96cm, 95% CI: −1.84 to −0.08, *p* = 0.0367), PFM (MD = −0.48%, 95% CI: −0.86 to 0.1, *p* = 0.0135), and VO_2peak_ (SMD = 0.19, 95% CI: 0.03 to 0.34, *p* = 0.0211). No statistical differences were found in BM (MD = −0.32 kg, 95% CI: −0.86 to −0.26, *p* = 0.2514), BMI (MD = 0.17 kg/m^2^, 95% CI: −0.11 to 0.46, *p* = 0.2511), FM (MD = −0.22 kg, 95% CI: −0.98 to −0.551, *p* = 0.5578), FFM (MD = −0.12 kg, 95% CI: −0.48 to 0.25, *p* = 0.5348), SBP (MD = 0.55 mmHg, 95% CI: −1.92 to 3.02, *p* = 0.6626), and DBP (MD = 0.68 mmHg, 95% CI: −0.76 to 2.13, *p* = 0.3523).

### 3.6. Subgroup Analysis

According to the different characteristics of included studies, studies were divided into 10 subgroups. The subgroup analysis was only performed on the between-group effect (HIIT vs. MICT). As the result showed, regardless of age, complication, duration, frequency, and HIIT interval, there were no statistical differences between HIIT and MICT on BM, FFM, SBP, and DBP. With regards to BMI, a significant effect of complications (obesity vs. other chronic diseases) was found in the subgroup analysis. The result indicated a significant effect of age (18–45 vs. 45–60 years), complications (obesity vs. other chronic diseases), and frequency (≤3 and >3 times/week) on WC. With regards to PFM, HIIT had a more significant effect on people who are 18–45 years or obese than MICT, and HIIT of >6 weeks or >3 times/week had a greater effect than MICT. Subgroup analysis of VO_2peak_ identified a significant effect of age, frequency, and HIIT interval (1–3 vs. ≥ 3 min) between HIIT and MICT (Table 4).

### 3.7. Sensitivity Analysis and Publication Bias

Low heterogeneity was detected in between-group analyses of VO_2peak_ (I^2^ = 9%), SBP (I^2^ = 26%), and DBP (I^2^ = 2%). After performing a sensitivity analysis by removing each one of the eligible studies, we found that the heterogeneity is due to the special forms of HIIT in two studies [39,44]. The results of funnel plots and Egger’s tests indicated no indication of publication bias.

## 4. Discussion

To our knowledge, this is the first systematic review to compare the intervention effectiveness of HIIT and MICT on body composition and CRF focusing on the young and middle-aged. There were 29 studies involving 807 participants (404 HIIT and 403 MICT) who were young and middle-aged (age from 18 to 65 years) combined in analyses. As the results showed, both HIIT and MICT caused improvements in BM, BMI, WC, PFM, FM, VO_2peak_, SBP, and DBP, and in the absence of significant influence in FFM. Notwithstanding, through data analysis, we did not find significant differences between HIIT and MICT on BM, BMI, FM, FFM, SBP, and DBP in the young and middle-aged, which was similar to the previous analysis [38,70]. However, HIIT was found to be superior to MICT in improving WC, PFM, and VO_2peak_. As the subgroup analysis indicated, there were statistical differences between HIIT and MICT on WC, PFM, and VO_2peak_ in the young, while these differences were not found in the middle-aged. We also found that HIIT is better at reducing BMI in people with other chronic diseases than MICT, and HIIT is a better choice for improving WC and PFM for people with obesity. Given intervention characteristics, compared to MICT, HIIT >3 times/week and >6 weeks might bring more positive influences on WC and PFM in the young and middle-aged respectively, and ≤3 times/week HIIT is more meaningful on VO_2peak_. HIIT of ≥3 min intervals seemed to cause more reduction in PFM, while HIIT of 1–3 min seemed to promote VO_2peak_ more. These findings might be of great help in designing strategies to improve body composition and CRF in the young and middle-aged. 

Aerobic exercise is considered to be the preferred exercise method for weight loss [71] and, as types of aerobic exercise, both HIIT and MICT can achieve meaningful reductions in BM and BMI. Although there were no significant differences found between HIIT and MICT on BM and BMI in the young and middle-aged, the within-group analyses showed that, despite the reduction being small (−2 kg and −0.9 kg/m^2^ for HIIT, and −2.19 kg and −0.92 kg/m^2^ for MICT), both interventions led to significant changes. Such results might be due to HIIT and MICT having similar influences on appetite [72] and sleep quality [73], which were important factors in BM reduction [74]. These findings are similar to the result of a recent network meta-analysis, with the author suggesting that exercise combined with a low-calorie diet might be more effective for weight loss than exercise alone [75]. A fasting plan is a good intervention to incorporate if people undergoing aerobic exercise want to achieve a greater improvement in BM and BMI [76]. In addition, diet composition is also an important extrinsic factor, as different diets were proven to have influences on the effectiveness of HIIT [77].

Compared to BM and BMI, abdominal adiposity is a more intuitive manifestation of visceral obesity, which is considered to be related to cardiometabolic risk [78] and all-cause mortality [79]. As a measure of abdominal adiposity, the increases in WC are often recognized to be associated with increases in visceral obesity [80]. It is evident from the results that both HIIT and MICT cause a meaningful reduction (>2 cm) in WC in the young and middle-aged, and that HIIT was superior to MICT, a finding consistent with a previous systematic review [81]. The meaningful improvement that HIIT brings may come from its positive effects on visceral adipose tissue [82]. Increased secretion of catecholamines which HIIT brings stimulates the β-adrenoceptors in the abdomen, causing the WC reduction [83]. From the subgroup analyses, we suggested that HIIT might be a better form of aerobic exercise for the young and people with obesity to improve WC than MICT, which needs further research to explore. In addition, only >3 days/week HIIT was found to be statistically different from MICT, which indicated that frequency might be more important than duration and HIIT interval in WC reduction. However, given that only four studies were involved in the >3 days/week subgroup, this result needs to be interpreted with caution. Despite no statistical differences being found in most subgroups between HIIT and MICT, HIIT had nearly 1cm more WC reduction on average than MICT. A similar superiority was also indicated in the older [84], hence we suggest that HIIT is a better exercise prescription for reducing WC than MICT.

As an important indicator of obesity, PFM has been proven to be independently related to reduced survival [85] and incidence of CVD [86] in the middle-aged and has been used in predictions of sports performance [87]. Our results indicated that both HIIT and MICT reduce PFM significantly (−2.03% for HIIT, and −1.89% for MICT) in the young and middle-aged, and HIIT leads to a −0.48% more significant reduction than MICT, which is inconsistent with a previous meta-analysis [39]. As one review [88] revealed, HIIT can achieve whole-body PFM reduction through increasing aerobic and anaerobic fitness, lowering insulin resistance, and increasing the skeletal muscle capacity for fatty acid oxidation and glycolytic enzyme content. These may be an explanation for our results. Our subgroup analyses found that HIIT is a more time-saving and effective prescription than MICT for the young (18–45 years) and obese (BMI ≥ 30 kg/m^2^). Moreover, we found that HIIT of >6 weeks and 3 min intervals is more superior to improve PFM in the young and middle-aged than MICT, which was contradictory to previous analyses [37,38,39]. Owing to the differences between our previous findings and the clinically meaningless reduction found in the results (<5% reduction [89]), although there is a consensus to achieve the same PFM reduction, HIIT consumes less time, whether HIIT is better for PFM reduction than MICT needs more studies, particularly on age, to find out.

Several studies have indicated that excess FM is a high-risk factor for all-cause mortality, high FFM is protection against mortality risk, and both are important predictors of functional outcomes and cardiometabolic diseases [90,91,92]. Similar to the results of recent studies, the within-group analyses revealed that HIIT and MICT achieve a significant decrease in the FM of the young and middle-aged. With regards to FFM, no positive effects were found in both the HIIT and MICT groups, as the results of a network meta-analysis [75] indicated that resistance training might be the best exercise form for improving FFM [93]. Although our results showed that HIIT is more significant in PFM reduction than MICT (even though the improvement was small), no statistical differences were found between HIIT and MICT in FM and FFM improvement in the participants. These findings were also revealed in a recent study that compared whole-body HIIT with traditional aerobic training. Despite whole-body HIIT being better for musculoskeletal improvement, no differences were found for fat mass or fat-free mass [94]. HIIT stimulates lipolysis through increasing catecholamines and growth hormone [95], while MICT has a greater proportion of fat as a substrate with a sustained high release of free fatty acids. Therefore, HIIT can bring more potential for muscle glycogen depletion than MICT. Subgroup analyses indicated that the effects of HIIT and MICT on postexercise fat and skeletal muscle oxidation are similar, regardless of the participants and exercise characteristics. At least, HIIT is more timesaving than MICT despite not having a greater improvement on FM and FFM.

High levels of CRF are proven to have benefits in reducing CVD and coronary heart disease (CHD) risk factors [96], and as the gold standard for CRF, maintaining or increasing VO_2peak_ is related to a decrease in incident hypertension risks [97]. Shreds of evidence confirmed that there is a positive association between flow-mediated dilation and CRF [98,99], hence we went one step further by including VO_2peak_, SBP, and DBP in the analysis. The results of the within-group analysis are unsurprising given the ability of HIIT and MICT to improve CRF and blood pressure [100,101]. Furthermore, the between-group analysis demonstrated that, relative to MICT, HIIT brings more benefits in VO_2peak_, which is similar to a recent study [102], and these benefits are probably due to age, frequency, and HIIT interval. These may be due to HIIT provoking greater nitric oxide bioavailability which MICT cannot [103]. More benefits HIIT brings in brachial artery flow-mediated dilation and mitochondrial function in the lateral vastus muscle may also make HIIT better than MICT in improving CRF [104]. In contrast, no special factors were found that can engender statistical differences between HIIT and MICT in SBP and DBP in this review. For young people who can only maintain a low frequency (≤3 times/week) in exercise, HIIT might be both a timesaving and clinically meaningful exercise prescription. The advantages that HIIT brings to blood flow through supplying oxygen to the muscles might not make it significantly different from MICT in improving vascular function, which is inconsistent with previous studies [70].

Although we conducted a comprehensive study on the effect of HIIT and MICT on fat loss and cardiorespiratory fitness in the young and middle-aged through meta-analysis and subgroup-analysis, this review still has several limitations. Firstly, the biggest limitation is that the participants we included in our analysis had several diseases (ex: obesity, diabetes, and fibromyalgia), which made the results insufficiently scientific. The results of this meta-analysis need to be carefully applied. More research needs to be carried out to find out the differences between the effects of HIIT and MICT in patients with specific diseases (ex: obesity, diabetes, and so on). In addition, as a result of strict inclusion and exclusion criteria, the subjects of all the eligible studies were limited (<30) and most of them (n = 20) were people with obesity which makes the results of our study lack universal applicability to this age group. Secondly, the mean dropout rate of HIIT and MICT were 9.76% and 7.72% respectively, with HIIT’s rate being a bit higher. Given that HIIT protocols are hard to tolerate by inactive people [105], supervision might be necessary to guarantee the implementation of HIIT. Although both laboratory-HIIT (supervised) and home-HIIT (unsupervised) were proven to have a meaningful improvement on CRF [106], a recent study [37] indicated that supervision of exercise can positively improve the effects of HIIT. Since almost all the included studies in our analysis were supervised, we did not include it as a covariable. Whether supervision is important in the effectiveness of HIIT needs more evidence to prove. In addition, the dietary control of participants was incorrectly analyzed in this study which might influence the outcomes, as several studies have proven that the combination of exercise and diet intervention has more effects on body composition [107,108]. Finally, notwithstanding statistical differences found in this meta-analysis, relative to MICT, the improvement HIIT brings was so limited that whether HIIT has more clinical meaning on fat loss and CRF is hard to say. The low time cost makes HIIT a more suitable exercise prescription for the young and middle-aged to apply when it comes to fat loss and CRF improvement. Moreover, HIIT has greater exercise adherence [109] and brings more exercise enjoyment [110,111] than MICT, which makes it a better plan to improve body composition and CRF for the young and middle-aged. However, a limited short-term (<6 months) improvement compared to MICT and the potential risk of a sudden high exercise intensity require clinicians to be careful when applying these results in practice. Future studies must pay more attention to the forms of HIIT and the combination with dietary intervention to expand the clinical significance of HIIT in fat loss and CRF improvement. Studies with a larger sample size, longer interventions, and better assessments need to be conducted to provide more compelling evidence to elucidate the timesaving and efficiency of HIIT.

## 5. Conclusions

Both HIIT and MICT appear to have a significant improvement on indicators of body composition and CRF, excluding FFM, in the young and middle-aged. HIIT provides more benefits on WC, PFM, and VO_2peak_ relative to MICT, which might be influenced by many factors, including age (18–45 years), complications (obesity), duration (>6 weeks), frequency, and HIIT interval. For the young and middle-aged, HIIT can achieve more improvement in abdominal obesity and aerobic ability than MICT. In summary, compared to MICT, our study indicated that the advantages HIIT brings to the young and middle-aged on fat loss and CRF are limited, yet these benefits can be provoked in a more time-saving manner.

## Figures and Tables

**Figure 1 ijerph-20-04741-f001:**
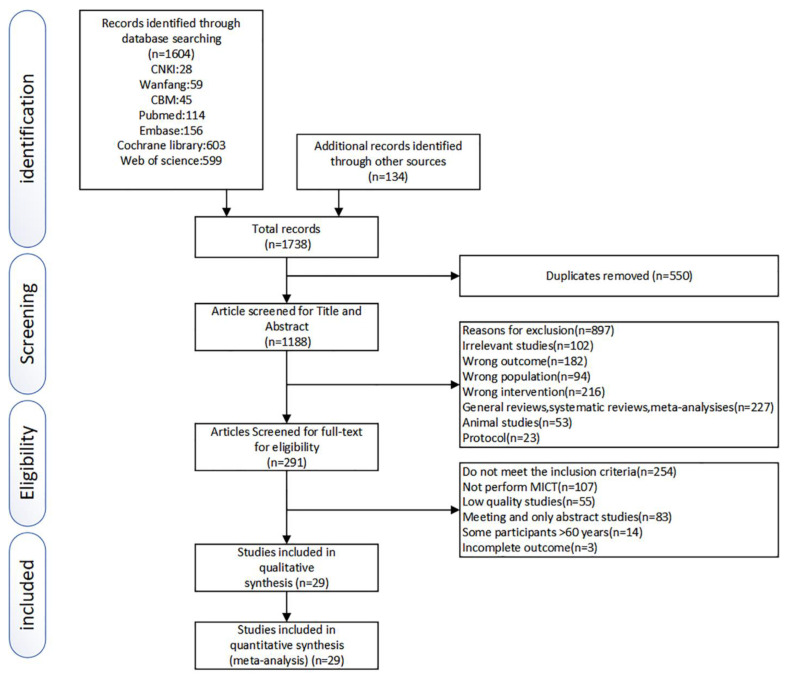
Literature search and study selection process.

**Figure 2 ijerph-20-04741-f002:**
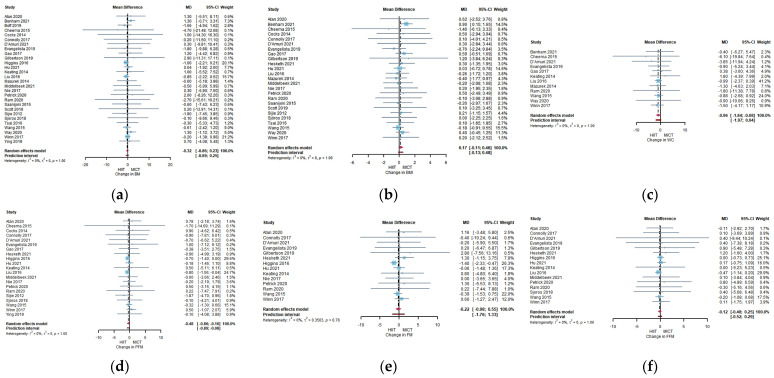
Forest plot for between-group effects of HIIT and MICT. (**a**) Forest plot for between-group effects of HIIT and MICT on body mass(BM); (**b**) Forest plot for between-group effects of HIIT and MICT on body mass index(BMI); (**c**) Forest plot for between-group effects of HIIT and MICT on waist circumference (WC); (**d**) Forest plot for between-group effects of HIIT and MICT on percent fat mass (PFM); (**e**) Forest plot for between-group effects of HIIT and MICT on fat mass (FM); (**f**) Forest plot for between-group effects of HIIT and MICT on fat-free mass (FFM); (**g**) Forest plot for between-group effects of HIIT and MICT on VO_2peak_; (**h**) Forest plot for between-group effects of HIIT and MICT on systolic blood pressure(SBP); (**i**) Forest plot for between-group effects of HIIT and MICT on diastolic blood pressure(DBP) [41,42,43,44,45,46,47,48,49,50,51,52,53,54,55,56,57,58,59,60,61,62,63,64,65,66,67,68,69].

**Table 1 ijerph-20-04741-t001:** Criteria for inclusion and exclusion.

PICOS	Inclusion	Exclusion
Participant	young and middle-aged (18–60 years old)	Age-incompatible
Intervention	HIIT	Intensity < 80–100% VO_2peak_ or HRmax; Other intervention
Comparison	MICT	Duration < 15 min; Other intervention
Outcome	BM; BMI; WC; PFM; FM; FFM; VO_2peak_; SBP; DBP	Other outcomes
Study	RCT	Books; opinion articles; observational studies; reviews; prospective cohort studies; studies and abstracts without adequate data

BM, body mass; BMI, body mass index; DBP, diastolic blood pressure FM, fat mass; FFM, fat-free mass; PFM, percent fat mass; SBP, Systolic blood pressure; RCT, randomized controlled trials; WC, waist circumference.

**Table 2 ijerph-20-04741-t002:** Characteristics of included studies.

Study	Duration	Age (Years)	Mela/ Female or Total	Participant	Types of Sport	HIIT					MICT				
						Sample Size	Exercise Intensity	Exercise Time per Times	Frequency	Dropouts (Attendance Rate)	Sample Size	Exercise Intensity	Exercise Time per Times	Frequency	Dropouts (Attendance Rate)
Atan 2020 [41]	6 weeks	47.6 ± 8.7	0/40	Fibromyalgia	Bicycle	19	4 min 80–95% HRmax; 3 min 70% HRmax	25 min	5 times/week	1 (95%)	19	65–70% HRmax	45 min	5 times/week	1 (95%)
Benham 2021 [42]	6 months	18–40	0/30	Polycystic ovary syndrome	Bicycle	16	30 s 90% HRmax; 90 s low-intensity aerobic exercise	20 min	3 times/week	0	14	50–60% HRmax	40 min	3 times/week	0
Boff 2019 [43]	2 months	23.5 ± 6	8/10	Type 1 diabete	Bicycle	9	1 min 80–85% HRmax; 4 min 50% HRmax	30 min	3 times/week	0	9	60–65% HRmax	20–30 min	3 times/week	0
Cheema 2015 [44]	3 months	39 ± 17	5/7	Sedentary obesity	Boxing/walking	6	2 min 15–20 RPE; 1 min passive recovery	30 min	1 times/week	0	6	4 MET	45 min	1 times/week	0
Cocks 2014 [45]	4 weeks	25 ± 1	16/0	Sedentary obesity	Bicycle	8	30 s 200% Wmax; 2 min 30 W for recovery	10–15 min	5 times/week	0	8	65% VO2peak	40–60 min	5 times/week	0
Connolly 2017 [46]	3 months	43.5 ± 7	0/36	Sedentary obesity	Bicycle	15	30 s 30% maximum effort; 20 s 50–60% maximum effort; 10 s 90% maximum effort	15–25 min	3 times/week	1 (94%)	15	A self-paced intensity	30–50 min	3 times/week	1 (94%)
D’Amuri 2021 [47]	3 months	38.7 ± 8.1	17/15	Sedentary obesity	Running	16	3 min 100% VO2peak; 1.5 min 50% VO2peak	13.5–31.5 min	3 times/week	0	16	60% VO2peak	30 min	3 times/week	0
Evangelista 2019 [48]	6 weeks	28.5 ± 5.6	25	Sedentary obesity	Whole body HIIT/Running	14	30 s all-out exercise; 30 s passive recovery	20 min	3 times/week	0	11	80% HRmax	20 min	3 times/week	0
Gao 2017 [49]	3 months	21.6 ± 1.4	25/25	Sedentary obesity	Running	17	4min 85% VO2peak; 2 min 50% VO2peak	30 min	5 times/week	8 (67%)	17	60% VO2peak	40 min	5 times/week	8 (67%)
Gilbertson 2019 [50]	4 months	48.3 ± 4.4	29	Prediabetes	Running	6	30 s 19–20 RPE; 4 min active rest	18–45 min	3 times/week	11 (35%)	9	45–55% HRmax	30–60 min	3 times/week	3 (75%)
Hesketh 2021 [51]	3 months	48 ± 10	88/66	Sedentary obesity	Home-based HIIT/ Running	21	1 min ≥ 80 HRmax; 1 min passive recovery	8–18 min	3 times/week	66 (24%)	29	50–70% HRmax	15–45 min	3 times/week	38 (43%)
Higgins 2016 [52]	6 weeks	20.4 ± 1.5	0/52	Sedentary obesity	Bicycle	23	30 s all-out exercise; 4 min active recovery	22.5–31.5 min	3 times/week	0	29	60–70% HRmax	20–30 min	3 times/week	0
Hu 2021 [53]	3 months	21.1 ± 1.4	0/33	Sedentary obesity	Bicycle	15	4 min 90% VO2peak; 3 min passive recovery	28–54 min	3 times/week	2 (88%)	15	60% VO2peak	40–60 min	3 times/week	1 (89%)
Keating 2014 [54]	3 months	43 ± 8.3	5/21	Sedentary obesity	Bicycle	11	30–45 s 120% VO2peak; 2–3 min low intensity	14–18 min	3 times/week	2 (85%)	11	50–65% VO2peak	30–42 min	3 times/week	2 (85%)
Liu 2016 [55]	3 months	20–23	0/40	Sedentary obesity	Bicycle	20	1 min 90% VO2peak; 1 min 20% VO2peak	30 min	4 times/week	0	20	50% VO2peak	30 min	4 times/week	0
Mazurek 2014 [56]	2 months	19.5 ± 0.6	0/46	Sedentary	Bicycle	24	10 s maximal sprinting 1 min 65–75% HRmax	32 min	3 times/week	0	22	65–75% HRmax	32 min	3 times/week	0
Middelbeek 2021 [57]	2 weeks	48 ± 5	22/0	Sedentary obesity	Bicycle	12	30 s all-out exercise; 4 min passive recovery	27 min	3 times/week	0	10	60% VO2peak	40–60 min	3 times/week	0
Nie 2017 [58]	3 months	21 ± 1.4	0/32	Sedentary obesity	Bicycle	16	4 min 90% VO2peak; 3 min passive recovery	300 kJ	3–4 times/week	1 (88%)	14	60% VO2peak	300 kJ	3–4 times/week	1 (93%)
Petrick 2020 [59]	6 weeks	37.4 ± 15.1	23/0	Sedentary obesity	Bicycle	12	30 s 170% Wpeak; 2 min passive recovery	10–15 min	3 times/week	0	11	60% Wpeak	30–40 min	5 times/week	0
Ram 2020 [60]	6 weeks	28 ± 7	28/0	Sedentary obesity	Bicycle	16	1 min 90% HRmax; 1 min 15% Wpeak	20 min	3 times/week	0	12	65–75% HRmax	30 min	3 times/week	0
Saanijoki 2015 [61]	2 weeks	48 ± 5	26/0	Sedentary obesity	Bicycle	13	30 s 180% peak workload sprints; 4min passive recovery	18–27 min	3 times/week	1 (93%)	13	60% peak workload	40–60 min	3 times/week	1 (93%)
Scott 2019 [62]	6 weeks	29 ± 10.6	10/4	Type 1 diabete	Bicycle	7	1 min 100% VO2peak; 1 min recovery at 50W	12–20 min	3 times/week	0	7	65% VO2peak	30–50 min	3 times/week	0
Sijie 2012 [63]	3 months	19.6 ± 0.8	0/40	Sedentary obesity	Running	17	3 min 85% VO2peak; 3 min 50% VO2peak	30 min	5 times/week	3 (85%)	16	50% VO2peak	40 min	5 times/week	4 (80%)
Sjöros 2018 [64]	2 weeks	49 ± 4	16/10	Type 2 diabete or prediabetes	Bicycle	11	30 s all-out exercise; 4 min passive recovery	18–27 min	3 times/week	2 (85%)	10	60% VO2peak	40–60 min	3 times/week	3 (77%)
Tsai 2016 [65]	6 weeks	22.3 ± 5.9	40/0	Sedentary	Bicycle	20	3 min 80% VO2peak; 3 min 40% VO2peak	30 min	5 times/week	0	20	60% VO2peak	30 min	5 times/week	0
Wang 2015 [66]	3 months	20.8 ± 1.1	0/24	Sedentary obesity	Running	12	4min 85–95% HRmax; 7 min 50–60% HRmax	44 min	4 times/week	0	12	60–70% HRmax	33 min	4 times/week	0
Way 2020 [67]	3 months	55.9 ± 2.3	26	Type 2 diabete	Bicycle	12	4 min 90% VO2peak; 5 min 50% VO2peak	9 min	3 times/week	0	12	60% VO2peak	45 min	3 times/week	2 (86%)
Winn 2017 [68]	4 weeks	43.5 ± 11.5	18	Sedentary obesity	Bicycle	8	4 min 80% VO2peak; 3 min 50% VO2peak	400 kJ	4 times/week	1 (89%)	8	55% VO2peak	400 kJ	4 times/week	1 (89%)
Ying 2019 [69]	3 months	35–45	18/0	Sedentary obesity	Running	8	2 min 90% HRmax; 1 min passive recovery	21 min	3 times/week	1 (89%)	8	65–70% HRmax	40 min	3 times/week	1 (89%)

HIIT, high-intensity interval training; HRmax, heart rate maximum; MICT, moderate-intensity continuous training; MET, metabolic equivalent RPE, Rating of Perceived Exertion; VO_2peak_, peak aerobic capacity; W, watts.

**Table 3 ijerph-20-04741-t003:** Details of meta-analysis.

Outcome		Within-Group Effects						Between-Group Effects				
	Included Studies (n)	HIIT			MICT			HIIT vs. MICT			Heterogeneity	
		n	MD	*p*	n	MD	*p*	MD	95% CI	*p*	I^2^ (%)	*p*
BM (kg)	28	383	−2	**0.0019 ***	374	−2.19	**0.0011 ***	−0.32	−0.86 to −0.26	0.2514	0	1
BMI (kg/m^2^)	25	353	−0.9	**0.000 ***	346	−0.92	**0.000 ***	0.17	−0.11 to 0.46	0.2511	0	1
WC (cm)	12	172	−4.41	**0.002 ***	161	−2.96	**0.000 ***	−0.96	−1.84 to −0.08	**0.0367**	0	1
PFM (%)	21	276	−2.03 #	**0.000 ***	286	−1.89 #	**0.000 ***	−0.48	−0.86 to −0.1	**0.0135**	0	1
FM (kg)	14	183	−1.79 #	**0.0028 ***	182	−2.33 #	**0.0002 ***	−0.22	−0.98 to 0.55	0.5578	0	0.78
FFM (kg)	16	210	−0.36 #	0.4867	208	−0.38 #	0.4577	−0.12	−0.48 to 0.25	0.5348	0	1
VO_2peak_	27	371	0.83 $	**0.000 ***	373	0.6 $	**0.000 ***	0.19 $	0.03 to 0.34	**0.0211**	9	0.33
SBP (mmHg)	14	166	−3.83	**0.0097 ***	165	−3.56	**0.002 ***	0.55	−1.92 to 3.02	0.6626	26	0.17
DBP (mmHg)	14	166	−1.59	**0.034**	165	−1.88	**0.024**	0.68	−0.76 to 2.13	0.3523	2	0.96

Bold indicates significant change (*p* < 0.05); * indicates significant heterogeneity(*p* < 0.01); # indicates data missing from 1 study, not included in analysis; $ indicates SMD instead of MD. BM, body mass; BMI, body mass index; CI, confidence intervals; DBP, diastolic blood pressure; FM, fat mass; FFM, fat-free mass; HIIT, high-intensity interval training; MICT, moderate-intensity continuous training; MD, mean difference; PFM, percent fat mass; SMD, standardized mean difference; SBP, systolic blood pressure; WC, waist circumference.

**Table 4 ijerph-20-04741-t004:** Summary of HIIT vs. MICT subgroup meta-analysis.

Outcome	Studies (n)	MD	(95% CI)	*p*	Heterogeneity	
					I^2^ (%)	*p*
BM (kg)						
Age: 18–45 years	21	MD = 0.19	−0.72 to 1.1	0.6841	0%	0.9993
Age: 45–60 years	7	MD = −0.46	−1.25 to 0.32	0.249	0%	0.9819
Complications: obesity	19	MD = −0.47	−1.15 to 0.22	0.1846	0%	0.9938
Complications: other chronic disease	7	MD = −0.31	−1.26 to 0.65	0.5321	0%	0.9985
Duration: ≤6 weeks	12	MD = 0.02	−1.03 to 1.06	0.9723	0%	0.9954
Duration: >6 weeks	16	MD = −0.44	−1.08 to 0.19	0.1727	0%	0.981
Frequency: ≤3 times/week	19	MD = −0.53	−1.14 to 0.07	0.0829	0%	1
Frequency: >3 times/week	9	MD = 0.62	−0.64 to 1.88	0.3348	0%	0.9098
HIIT interval: 1−3 min	15	MD = −0.1	−0.93 to 0.73	0.8124	0%	0.9995
HIIT interval: ≥3 min	13	MD = −0.47	−1.21 to 0.27	0.2093	0%	0.9239
BMI (kg/m^2^)						
Age: 18−45 years	18	MD = 0.05	−0.24 to 0.33	0.7494	0%	0.5159
Age: 45−60 years	7	MD = 0.28	−0.1 to 0.66	0.1525	0%	0.9916
Complications: obesity	17	MD = 0.12	−0.12 to 035	0.3427	0%	0.9971
Complications: other chronic disease	6	MD = 0.63	0.1 to 1.16	**0.0189**	0%	0.9426
Duration: ≤6 weeks	11	MD = 0.11	−0.26 to 0.49	0.5575	0%	0.9983
Duration: >6 weeks	14	MD = 0.12	−0.18 to 0.42	0.4225	29%	0.1459
Frequency: ≤3 times/week	16	MD = 0.26	−0.04 to 0.55	0.0855	0%	0.9608
Frequency: >3 times/week	9	MD = −0.03	−0.33 to 0.27	0.8313	0%	0.5479
HIIT interval: 1−3 min	12	MD = 0.12	−0.17 to 0.41	0.4084	0%	0.9991
HIIT interval: ≥3 min	13	MD = 0.12	−0.28 to 0.53	0.5521	27%	0.1711
WC (cm)						
Age: 18−45 years	11	MD = −0.96	−1.85 to −0.07	**0.0338**	0%	0.9909
Age: 45−60 years	1	MD = −0.9	−10.06 to 8.26	NA	NA	NA
Complications: obesity	9	MD = −0.95	−1.88 to −0.02	**0.0461**	0%	0.9652
Complications: other chronic disease	2	MD = −0.55	−5.49 to 4.4	0.8288	0%	0.9283
Duration: ≤6 weeks	3	MD = −1.36	−3.58 to 0.86	0.229	0%	0.9696
Duration: >6 weeks	9	MD = −0.89	−1.85 to 0.08	0.0717	0%	0.9709
Frequency: ≤3 times/week	8	MD = −1.03	−3.07 to 0.99	0.3164	0%	0.9657
Frequency: >3 times/week	4	MD = −0.94	−1.92 to −0.08	**0.0397**	0%	0.8962
HIIT interval: 1−3 min	4	MD = −0.92	−2.35 to 0.52	0.2113	0%	0.8198
HIIT interval: ≥3 min	8	MD = −0.99	−2.11 to 0.13	0.0839	0%	0.9798
PFM (%)						
Age: 18−45 years	16	MD = −0.5	−0.89 to −0.11	**0.0112**	0%	0.9969
Age: 45−60 years	4	MD = −0.1	−1.82 to 1.61	0.9074	0%	0.8989
Complications: obesity	18	MD = −0.54	−0.94 to −0.14	**0.0085**	0%	0.9989
Complications: other chronic disease	2	MD = 0.48	−1.92 to 2.88	0.6953	0%	0.7335
Duration: ≤6 weeks	9	MD = −0.01	−0.89 to 0.88	0.995	0%	0.9844
Duration: >6 weeks	11	MD = −0.62	−1.15 to −0.08	**0.0252**	0%	0.9977
Frequency: ≤3 times/week	12	MD = −0.61	−1.24 to 0.01	0.0576	0%	1
Frequency: >3 times/week	8	MD = −0.45	−0.96 to 0.06	0.086	0%	0.7558
HIIT interval: 1−3 min	9	MD = −0.43	−0.92 to 0.06	0.0823	0%	0.8745
HIIT interval: ≥3 min	11	MD = −0.67	−1.36 to −0.06	**0.0276**	0%	0.9996
FM (kg)						
Age: 18−45 years	11	MD = −0.45	−1.2 to 0.3	0.2443	0%	0.8256
Age: 45−60 years	3	MD = 1.33	−0.79 to 3.45	0.2174	0%	0.9593
Complications: obesity	12	MD = −0.27	−1.04 to 0.49	0.485	0%	0.7089
Complications: other chronic disease	2	MD = 1.43	−2.8 to 5.67	0.507	0%	0.7771
Duration: ≤6 weeks	6	MD = −0.32	−1.8 to 1.16	0.6695	0%	0.4261
Duration: >6 weeks	8	MD = −0.05	−0.84 to 0.75	0.9105	0%	0.9652
Frequency: ≤3 times/week	10	MD = −0.25	−1.4 to 0.9	0.666	0%	0.6651
Frequency: >3 times/week	4	MD = −0.06	−0.99 to 0.86	0.8909	0%	0.7852
HIIT interval: 1−3 min	8	MD = −0.41	−1.19 to 0.37	0.3026	0%	0.4914
HIIT interval: ≥3 min	6	MD = 0.9	−1.03 to 2.83	0.3587	0%	0.9954
FFM (kg)						
Age: 18−45 years	11	MD = −0.15	−0.52 to 0.23	0.4425	0%	0.9979
Age: 45−60 years	5	MD = 0.48	−1.16 to 2.13	0.5663	0%	0.9758
Complications: obesity	13	MD = −0.12	−0.49 to 0.25	0.5208	0%	0.9976
Complications: other chronic disease	3	MD = 0.11	−2.26 to 2.47	0.9298	0%	0.9555
Duration: ≤6 weeks	8	MD = 0.02	−0.62 to 0.66	0.9475	0%	1
Duration: >6 weeks	8	MD = −0.18	−0.63 to 0.26	0.4209	0%	0.9374
Frequency: ≤3 times/week	12	MD = 0.12	−0.41 to 0.65	0.6578	0%	1
Frequency: >3 times/week	4	MD = −0.33	−0.83 to 0.18	0.2036	0%	0.9213
HIIT interval: 1−3 min	9	MD = 0	−0.45 to 0.45	0.9994	0%	0.9999
HIIT interval: ≥3 min	7	MD = −0.34	−0.97 to 0.29	0.2864	0%	0.9554
VO_2peak_						
Age: 18−45 years	20	SMD = 0.28	0.1 to 0.46	**0.0025**	9%	0.3316
Age: 45−60 years	7	SMD = −0.08	−0.36 to 0.21	0.5956	0%	0.8155
Complications: obesity	18	SMD = 0.16	−0.04 to 0.35	0.1084	11%	0.3231
Complications: other chronic disease	7	SMD = 0.12	−0.23 to 0.46	0.5061	10%	0.3521
Duration: ≤6 weeks	11	SMD = 0.17	−0.1 to 0.45	0.2271	22%	0.23
Duration: >6 weeks	16	SMD = 0.17	−0.01 to 0.36	0.0703	4%	0.4054
Frequency: ≤3 times/week	19	SMD = 0.24	0.03 to 0.44	**0.0219**	18%	0.2324
Frequency: >3 times/week	8	SMD = 0.08	−0.17 to 0.34	0.5196	0%	0.5765
HIIT interval: 1−3 min	15	SMD = 0.27	0.04 to 0.5	**0.0228**	24%	0.1807
HIIT interval: ≥3 min	12	SMD = 0.08	−0.14 to 0.29	0.4981	0%	0.702
SBP (mmHg)						
Age: 18−45 years	11	MD = 0.15	−3.32 to 3.62	0.9327	40%	0.0798
Age: 45−60 years	4	MD = 1.47	−2.31 to 5.25	0.4461	0%	0.6178
Complications: obesity	8	MD = −1.05	−4.78 to 2.67	0.5804	38%	0.1231
Complications: other chronic disease	6	MD = 3.27	−0.48 to 7.01	0.0873	0%	0.4631
Duration: ≤6 weeks	5	MD = 0.7	−3.76 to 5.17	0.7585	0%	0.9592
Duration: >6 weeks	10	MD = 0.13	−3.59 to 3.85	0.9447	50%	0.0333
Frequency: ≤3 times/week	11	MD = 0.15	−3.45 to 3.75	0.9353	45%	0.0513
Frequency: >3 times/week	4	MD = 0.65	−3.59 to 4.89	0.7645	0%	0.9025
HIIT interval: 1−3 min	7	MD = 1.88	−1.25 to 5.01	0.2397	0%	0.451
HIIT interval: ≥3 min	8	MD = −0.74	−4.82 to 3.34	0.721	41%	0.1018
DBP (mmHg)						
Age: 18−45 years	11	MD = 0.58	−1.22 to 2.39	0.5268	4%	0.4041
Age: 45−60 years	4	MD = 0.96	−1.89 to 3.81	0.5105	23%	0.2729
Complications: obesity	8	MD = 0.36	−1.39 to 2.11	0.6866	23%	0.2421
Complications: other chronic disease	6	MD = 1.64	−1.17 to 4.45	0.2517	0%	0.4985
Duration: ≤6 weeks	5	MD = 0.67	−2.53 to 3.87	0.683	0%	0.5807
Duration: >6 weeks	10	MD = 0.68	−0.93 to 2.29	0.4048	21%	0.2438
Frequency: ≤3 times/week	11	MD = 0.74	−1.15 to 2.64	0.4177	18%	0.2699
Frequency: >3 times/week	4	MD = 0.58	−1.75 to 2.92	0.6244	0%	0.55
HIIT interval: 1−3 min	7	MD = 1.31	−0.67 to 3.29	0.1956	0%	0.584
HIIT interval: ≥3 min	8	MD = −0.02	−2.11 to 2.07	0.9853	21%	0.2639

Bold indicates significant change (*p* < 0.05). BM, body mass; BMI, body mass index; CI, confidence HIIT intervals; DBP, diastolic blood pressure; FM, fat mass; FFM, fat-free mass; HIIT, high-intensity HIIT interval training; MICT, moderate-intensity continuous training; MD, mean difference; PFM, percent fat mass; SMD, standardized mean difference; SBP, Systolic blood pressure; WC, waist circumference.

## Data Availability

All the included studies are in Table 2.

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
