# Peer review of "Effect of High-Intensity Interval Training vs. Moderate-Intensity Continuous Training on Fat Loss and Cardiorespiratory Fitness in the Young and Middle-Aged a Systematic Review and Meta-Analysis"

_ijerph, 2023, doi:10.3390/ijerph20064741_

Round 1

Reviewer 1 Report

Thanks for letting me review your paper, however, the paper has one major concern that I think can not be amended.

The sample is not specific, so it might not be an accurate analysis. It's nonsense to combine the effect of exercise on a normal and diseased population (ex: PCO, DM) in one sample. This is not meaningful or useful for the readers. Also, might be not enough scientific.

Another concern that can be amended easily is the English language. In general, it's fine, however, on many occasions, it is hard for me to understand what is written. However, this is not a big concern.

I am sorry for that and wish you the best of luck in your future projects.

Author Response

Firstly all the authors would like to thank the reviewer. And We are ashamed of the major flaws and concerns in our article. We have tried our best to modify and improve. The response is shown in the attachment. Please see the attachment

Reviewer 2 Report

Effect of High-intensity interval training vs moderate-intensity continuous training on fat loss and cardiorespiratory fitness in the young and middle-aged a systematic review and meta-analysis

First of all, the reviewer would like to thank the authors for their work and efforts in trying to improve sports science knowledge.

General comments to the authors

This systematic review is conducted to evaluate the effect of high-intensity interval training (HIIT) and moderate-intensity continuous training (MICT) on body composition and cardiorespiratory fitness (CRF) in the young and middle-aged. Overall, the study is well-designed and well-written, with a great introduction proposing the usefulness of the topic and a clear outline of the research question. I suggest that the author modify/include some suggestions to improve the manuscript before be published:

Show references throughout the manuscript the right way. Please check and fix all references throughout the manuscript.

P 4 Line 131: RCT: randomized controlled trials should be added in Table 1.

P 4 Line 138: ??

P 4 Line 165: ??

The authors should add these important articles about high-intensity interval training and moderate-intensity continuous training to support their ideas.

Arslan, E., Can, S., & Demirkan, E. (2017). Effect of short-term aerobic and combined training program on body composition, lipids profile and psychological health in premenopausal women. Science & Sports, 32(2), 106-113.

Wisløff, U., Støylen, A., Loennechen, J. P., Bruvold, M., Rognmo, Ø., Haram, P. M., ... & Skjærpe, T. (2007). Superior cardiovascular effect of aerobic interval training versus moderate continuous training in heart failure patients: a randomized study. Circulation, 115(24), 3086-3094.

Bartlett, J. D., Close, G. L., MacLaren, D. P., Gregson, W., Drust, B., & Morton, J. P. (2011). High-intensity interval running is perceived to be more enjoyable than moderate-intensity continuous exercise: implications for exercise adherence. Journal of sports sciences, 29(6), 547-553.

Helgerud, J., Høydal, K., Wang, E., Karlsen, T., Berg, P., Bjerkaas, M., ... & Hoff, J. (2007). Aerobic high-intensity intervals improve VË™ O2max more than moderate training. Medicine & science in sports & exercise, 39(4), 665-671.

Martinez, N., Kilpatrick, M. W., Salomon, K., Jung, M. E., & Little, J. P. (2015). Affective and enjoyment responses to high-intensity interval training in overweight-to-obese and insufficiently active adults. Journal of Sport and Exercise Psychology, 37(2), 138-149.

Oliveira, B. R. R., Santos, T. M., Kilpatrick, M., Pires, F. O., & Deslandes, A. C. (2018). Affective and enjoyment responses in high-intensity interval training and continuous training: A systematic review and meta-analysis. PloS one, 13(6), e0197124.

Soylu, Y., Arslan, E., Sogut, M., Kilit, B., & Clemente, F. (2021). Effects of self-paced high-intensity interval training and moderate-intensity continuous training on the physical performance and psychophysiological responses in recreationally active young adults. Biology of Sport, 38(4), 555-562.

Oliveira, B. R., Slama, F. A., Deslandes, A. C., Furtado, E. S., & Santos, T. M. (2013). Continuous and high-intensity interval training: which promotes higher pleasure?. PloS one, 8(11), e79965.

Overall the discussion is well-written and incorporates relevant literature. The tables and figures sections are well-designed and well-written.

Author Response

Firstly thank you very much for your careful review and approval. Thank you very much for your careful review and approval. Please see the attachment.

Reviewer 3 Report

I believe the manuscript meets its purpose of systematically addressing the literature data on the effect of interval and high-intensity training (HIIT) and continuous medium-intensity training (MICT) on body composition and cardiorespiratory fitness , in young people and adults, by complying with the methodological precepts underlying a review of this kind, namely regarding the defining of the review question, the developing of the research strategy, the selecting of research sources and databases, the defining of inclusion and exclusion criteria, the screening and coding of studies, the assessment of their quality, and, finally, the synthesis and presentation of results.

*(page 4, 4th paragraph): Why were two softwares used? This option should be justified.  

*(page 18, 2nd paragraph): Why weren't there statistical differences between HIIT and MICT on WC, PFM and VO2 in middle-age? In case the response to this finding is not contemplated in literature, the Author´s should think and conjecture about possible causes that explain it.  

*(page 18, 3rd paragraph): Same issue as the previous comment. Could it be because diet issues, addictive behaviors that are suppressed, increased concerns about sleeping hours, etc.? Are there extrinsic factors that could explain this finding? Below is a reference that could help to discuss this issue:   

Maughan RJ, Greenhaff PL, Leiper JB, Ball D, Lambert CP, Gleeson M. Diet composition and the performance of high-intensity exercise. J Sports Sci. 1997 Jun;15(3):265-75. doi: 10.1080/026404197367272. PMID: 9232552.  

*(page 20, paragraph from the previous page): Why did your study not include this aspect as a covariable, considering the existing controversy concerning what your sentence states? As an example of that, please, check the reference below, where it is experimentally confirmed that HIIT done unsupervised at home may also be  a viable alternative to achieve good health standards.   Blackwell J, Atherton PJ, Smith K, Doleman B, Williams JP, Lund JN, Phillips BE. The efficacy of unsupervised home-based exercise regimens in comparison to supervised laboratory-based exercise training upon cardio-respiratory health facets. Physiol Rep. 2017 Sep;5(17):e13390. doi: 10.14814/phy2.13390. PMID: 28912129; PMCID: PMC5599857.  

Author Response

Firstly thank you very much for your careful review and approval. The response is shown in the attachment. Please see the attachment.
